# Species-Enriched Grass-Clover Mixtures Can Promote Bumblebee Abundance Compared with Intensively Managed Conventional Pastures

Henriette Beye [1,*], Friedhelm Taube [2,3], Katharina Lange [1], Mario Hasler [4], Christof Kluß [2], Ralf Loges [2] and Tim Diekötter [1]

1   Institute of Natural Resource Conservation, Landscape Ecology, Kiel University, 24118 Kiel, Germany; lange-ka@gmx.de (K.L.); tdiekoetter@ecology.uni-kiel.de (T.D.)
2   Institute of Crop Science and Plant Breeding, Grass and Forage Science/Organic Agriculture, Kiel University, 24118 Kiel, Germany; ftaube@gfo.uni-kiel.de (F.T.); ckluss@gfo.uni-kiel.de (C.K.); rloges@gfo.uni-kiel.de (R.L.)
3   Grass Based Dairy Systems, Animal Production Systems Group, Wageningen University (WUR), 6700 HB Wageningen, The Netherlands
4   Department of Statistics, Kiel University, 24118 Kiel, Germany; hasler@email.uni-kiel.de
*   Correspondence: hbeye@ecology.uni-kiel.de

**Abstract:** (1) Land use intensification has led to serious declines in biodiversity, including in forage production systems for dairy cows. Agri-environmental schemes, such as enriching grasslands in floral species, were shown to be an effective tool to promote biodiversity in higher trophic levels. Here, we studied an innovative pasture-based dairy production system with floral-species-enhanced temporary grasslands, with respect to the effect on wild bee abundance and species richness. (2) We studied three grass-clover mixtures with perennial ryegrass and clover species with different levels of plant diversity for flower cover and wild bees. The grass-clover pastures were rotationally stocked with cattle and parts of the pastures were excluded from grazing. Intensively managed conventional permanent grasslands were studied as the common land use type. Wild bees were caught by sweep netting. Wild bee diversity was analysed with a general linear mixed model. For species richness, an incidence-based coverage estimator was calculated. (3) In total, 541 wild bees from 10 species were found. No wild bees were caught on the conventional grasslands. Wild bee abundance and species richness did not differ among the three grass-clover mixtures, but with increasing flower cover of white clover (*Trifolium repens*), wild bee abundance increased. Except for one solitary wild bee individual, the recorded community exclusively consisted of bumblebees. While generalist species that are commonly found on farmland dominated, rare long-tongued bumblebees made up 10% on the grazed sites of the multispecies mixture and made up 20% on the ungrazed strips of the binary mixture and multispecies mixture. (4) We conclude that multispecies mixtures can provide resources for generalist bumblebee species, especially when compared with conventional grasslands that offer no resources. Considering that the multispecies mixture has been also shown to reduce greenhouse gas emissions and nitrate leaching, while maintaining high forage yields, our findings add to the potential to promote a wide range of ecosystem services. Yet, should their full potential be enfolded, grazing should partially be excluded, and the mixture should be extended by plant species with more open flowers, suitable for solitary wild bees.

**Keywords:** agri-environmental schemes; agrobiodiversity; flower-visiting insects; multifunctionality; pollinator decline; ecological farming; wild bees; ecological intensification; dairy systems; ley grassland

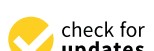

## 1. Introduction

The growing demand for food worldwide has fundamentally changed agricultural systems since the 1950s [1]. To meet this demand, land use has been intensified, resulting in

heavy use of synthetic fertilizers [2,3] and plant protection products [4] as well as increasing mechanization [5–7], with an associated fragmentation of seminatural habitats [8]. This intensification is driving spatial [9–11] and temporal [12] homogenization of agricultural landscapes, which imposes a serious threat for biodiversity and ecosystem functioning [13]. Thus, conservation and reintroduction of plant diversity is considered to be key in promoting agrobiodiversity [14,15]. In recent years, measures towards ecological intensification in agriculture have been discussed and promoted [16–18]. Here, we tested whether plant-species-enriched grass-clover mixtures in organically managed ley systems can provide such resources and promote wild bee diversity.

Flower-visiting insects, wild bees in particular, have an essential function in plant pollination [19,20] because wild plants and many agricultural crop species depend on successful pollination [21–23]. Pollination may be provided by few common species [24], but can be improved by species richness [25]. Increases in quantity [26], quality [27,28] and commercial value [28] of yield through insect pollination have been reported for many crops. While the proportion of agricultural crops that require biotic pollination has increased by more than 300% over the last 50 years [29], pollinating insects have simultaneously declined in their abundance [30] and diversity [31], and this trend has not yet been stopped.

Substantial declines in pollinators have been attributed to land use change [32], loss of habitats [33] or habitat quality [34], the effects of climate change [35], increased exposure to insecticides [36], the loss of plant resources [37] and higher susceptibility to diseases [38,39]. Landscape structure acts as a filter for pollinating species with specific functional traits, such as body size [40,41] or proboscis length [42,43], both among and within species. Consequently, not only wild bee abundance and species richness are affected by changes in landscape structure, with the functional composition of entire pollinator communities also being affected [40,44]. In order to maintain or restore the structural and functional diversity of wild bee communities and their benefits for yield quality and quantity, farming strategies that support insect diversity such as organic farming or agri-environmental schemes (AESs) need to be promoted and improved in their efficiency.

Measures to support pollinators aim to provide diverse nectar and pollen resources as well as nesting or overwintering sites. Among these measures, organic farming is a prominent one [45]. It has been shown to increase the diversity of plants, especially insect-pollinated plants, in arable land [46,47] but not necessarily in grassland [48]. Yet, in permanent grasslands of organic dairy farms, wild bee abundance was found to be higher than in conventional agriculture and found to be positively related to floral abundance, presumably white clover [49]. In addition to floral abundance, floral richness is also known to increase wild bee abundance and species richness [50,51]. As many grasslands in intensive agricultural areas are not permanent, species-enriched grass-clover mixtures seem like a promising option to promote wild bee diversity in ley systems. It has been shown previously that grass-legume-forb mixtures with enhanced botanical diversity and low grazing intensity can lead to increased functional diversity of pollinators compared with grass-based pastures [50,52,53]; however, the question remains whether this is feasible in intensively used ley systems.

In an organic ley system, we studied the effects of three grass-clover mixtures with varying levels of plant species richness and different management regimes on the abundance and richness of wild bee pollinators. We hypothesized that (1) organic grass-clover pastures promote wild bee abundance and species richness in comparison with conventional pastures, cut five times per year for silage; (2) wild bee abundance and species richness in organic grass-clover pastures increases with floral diversity; and (3) the potential of organic grass-clover pastures in promoting wild bee abundance and species richness is limited by intensive grazing.

## 2. Materials and Methods

### 2.1. Study Region and Design

The study took place in 2019 at the Lindhof experimental farm of Kiel University, in Schleswig-Holstein, Germany (WGS 84, 54° 28′ N; 9° 58′ E), in the frame of the topic "Eco-efficiency of pasture-based milk production" [54–56]. In 2019, the mean temperature of the study area was 10.24 °C (mean annual temperature of 9.35 °C between 1990 and 2019) and the annual precipitation was 745 mm (mean annual precipitation of 775 mm between 1990 and 2019). On the farm, a crop rotation system has been implemented since 2015. The rotational system began with grass-clover pastures that were sown in spring and then used as pastures for two consecutive years. In the third year, oat (*Avena sativa* L.) was sown, followed by potatoes and winter wheat. In the winter wheat crop, grass-clover leys were again established as understory in spring, ensuring a full productive stand already in August of the year of establishment. Grass-clover pastures were sown in mid-May of 2017 (pasture size of 4.4 ha for the binary mixture, 13.45 ha for the tertiary mixture, 6.35 ha for the multispecies mixture) and 2018 (pasture size of 3.4 ha for the binary mixture, 2.89 ha for the tertiary mixture, 3.1 ha for the multispecies mixture). For each mixture, there were two sites (one in the first and one in the second year of production) that we were able to sample. In 2019, we conducted field research when the pasture sown in 2017 was in the second year of production and the pasture sown in 2018 was in the first year of production. In total, 60.3 ha of grass-clover swards were present on the experimental farm in 2019. There were three different grass-clover mixtures: (1) the binary mixture contained perennial ryegrass (*Lolium perenne* L., 24 kg ha$^{-1}$) and white clover (*Trifolium repens* L., 4 kg ha$^{-1}$); (2) the tertiary mixture contained perennial ryegrass (24 kg ha$^{-1}$), white clover (2 kg ha$^{-1}$) and red clover (*Trifolium pratense* L., 6 kg ha$^{-1}$); and (3) the multispecies mixture contained perennial ryegrass (16 kg ha$^{-1}$), white clover (1.3 kg ha$^{-1}$), red clover (3 kg ha$^{-1}$) birdsfoot trefoil (*Lotus corniculatus* L., 3 kg ha$^{-1}$), chicory (*Cichorium intybus* L., 2 kg ha$^{-1}$), plantain (*Plantago lanceolata* L., 1 kg ha$^{-1}$), caraway (*Carum carvi* L., 2 kg ha$^{-1}$) and sheep's burnet (*Sanguisorba minor* L., 2 kg ha$^{-1}$). The varieties and details on composition are in Table A1. The grass-clover pastures were rotationally stocked every three or four weeks for two-four days from April to September (nine grazing events in total) with an average stocking rate of 2.0 livestock units per hectare. We treated the pastures from each mixture and each year as a single paddock. The cattle grazed on the paddocks in subsections that were measured for yield and divided according to the cattle's needs to allow allocation of fresh grass two times daily. Additionally, to evaluate the flower potential of the binary mixture and multispecies mixture, an area of 0.042 ha was excluded from grazing on each pasture of both mixtures. To minimize economic losses that follow the exclusion of pastures from grazing, we only implemented ungrazed strips on the two mixtures, not the tertiary mixture. Therefore, the resulting four grass-clover exclosures (two for the binary mixture and two for the multispecies mixture), were cut once on 20 August 2019 after all exclosures were sampled for wild bees, to simulate a less intensive usage where more flowers could develop, so that the maximum potential in flower cover of the grass-clover fields was displayed. In addition to the grass-clover mixtures, typical intensively managed conventional pastures (n = 3) with perennial ryegrass growing on the pastures and with five cuts during the season represented common grassland management. The pastures were cut on 22 May, 26 June, 7 August, 19 September and 29 October, and were fertilized on 4 March 2019 with 110 kg N ha$^{-1}$, on 27 May 2019 and on 1 July 2019 with 80 kg N ha$^{-1}$, and on 9 August of 2019 with 40 kg N ha$^{-1}$.

To assess yields on the grass-clover pastures, we took six random samples on each pasture in 0.25 m$^2$ squares and clipped vegetation to 5 cm residual height. The plots were sampled on nine dates before the grazing events. For the grazed pastures, we sampled the sites that were in the first and second year of production and calculated the mean of both years, as for the pollinator sampling. On the ungrazed strips of the grass-clover pastures, we also took six samples once before the mowing event and the sampling was repeated on five dates before each of the five mowing events on the conventional pastures. To estimate

the dry matter (DM) yields, the samples were dried for 48 h at 60 °C. We calculated the DM in tons per hectare and year.

### 2.2. Pollinator Monitoring

Pollinator abundance and diversity was recorded between 25 June and 13 September 2019, approximately two weeks after grazing events when flower cover was sufficient. Sweep netting was performed along three 50 m transects per grass-clover pasture, as well as one transect per grass-clover strip without grazing. Sweep netting was performed by walking the transects two times at a walking pace between 10 a.m. and 5 p.m. on sunny days with little wind on two occasions per transect [57]. Visited plant species were recorded for each wild bee caught [57]. Insects were stored in vials with 1–2 drops of ethyl acetate and frozen until species-level determination in the lab. Parallel to sweep netting, plant species richness and flower cover were recorded in 1 m$^2$ plots, randomly placed along the transect, where we estimated the coverage of blossoms in percent for each plant species. If the whole plot was covered with flower heads and no vegetation underneath was visible, this was considered to be 100% flower cover.

### 2.3. Statistical Analysis

We analysed the effect of grass-clover mixture (binary, tertiary and multispecies mixture) and management (grazed and ungrazed) on the wild bee abundance with a generalized linear mixed model with multivariate normal random effects, using penalized quasi-likelihood. For the data evaluation, we defined an appropriate generalized mixed model [58] for the abundance data and a mixed model [58] for the abundance data and a mixed model [59,60] for species diversity data [59,60]. For the latter, the ICE (incidence-based coverage estimator) index was calculated. The models incorporated a fixed pseudo factor, combining the actual factors mixture and management, which was necessary because the actual factors were not orthogonal [61]. The coverage of white clover was included in the model as a covariate. The experimental design was taken into account by the random factors transect, nested in management, nested in mixture and nested in the year of production of the mixture. The residuals of the abundance data and the ICE diversity data displayed a Poisson distribution. This assumption was established as a result of the graphical residual analysis. Based on these models, the pseudo r$^2$ was calculated [62]. Differences were investigated with an analysis of covariance (ANCOVA) function [63]. Next, multiple contrast tests [64,65] were performed to compare intercepts of the effects of mixture and management. All analyses were performed in R version 4.1.2 [66], using the vegan package [67].

## 3. Results

In total, we caught 541 wild bees belonging to 2 genera and 10 species. The most abundant species were *Bombus terrestris* (43.8%) and *B. lapidarius* (38.4%), followed by *B. lucorum* (6.5%). We recorded the bumblebee species *B. hortorum* (5%), *B. pascuorum* (4.3%), *B. bohemicus* (<1%), *B. hypnorum* (<1%), *B. pratorum* (<1%) and *B. soreensis* (<1%). Only one solitary wild bee, *Andrena flavipes,* was caught.

While we found an average flower cover of 18.4% across all grazed grass-clover pastures, no flower resources were present on the conventional grasslands. We caught 395 wild bees on the grazed organic grass-clover pastures, 146 wild bees on ungrazed organic grass-clover strips and no wild bees on the conventional grasslands. We observed bumblebees flying over the conventional grasslands; however, there were no flowering resources for them to visit.

### 3.1. Grass-Clover Mixtures

The flower cover varied among the three mixtures, but white clover was the dominating species in all mixtures. In the binary mixture, the overall flower cover was 19.2%, only composed of white clover. In the tertiary mixture, white clover flowers covered 7.9% and

red clover flowers covered 4.8% (overall flower cover 12.7%). In the multispecies mixture, we found 11.1% white clover flower cover, 4.3% red clover and 0.4% birdsfoot trefoil cover (overall flower cover 15.8%).

The grass-clover mixtures did not differ in regard to wild bee abundance (Table A2); although, the abundance of wild bees increased significantly with the flower cover of white clover ($p < 0.001$) in all three grass-clover mixtures and across all management types (Figure 1). There was a significant trend of an increase in the ICE diversity index with flower cover of white clover in all three mixtures and under all managements ($p = 0.062$, Figure 2), but, again, the mixtures did not differ in terms of wild bee diversity (Table A3). Although we did not find an effect of mixture on the ICE diversity index, we found 2.1% of the rare long-tongued bumblebee species on the grazed pastures of the binary mixture, 1.7% long-tongued bumblebees on the tertiary mixture and 10.3% long-tongued species on the multispecies mixture, thus indicating more long-tongued species in the mixture with most plant species.

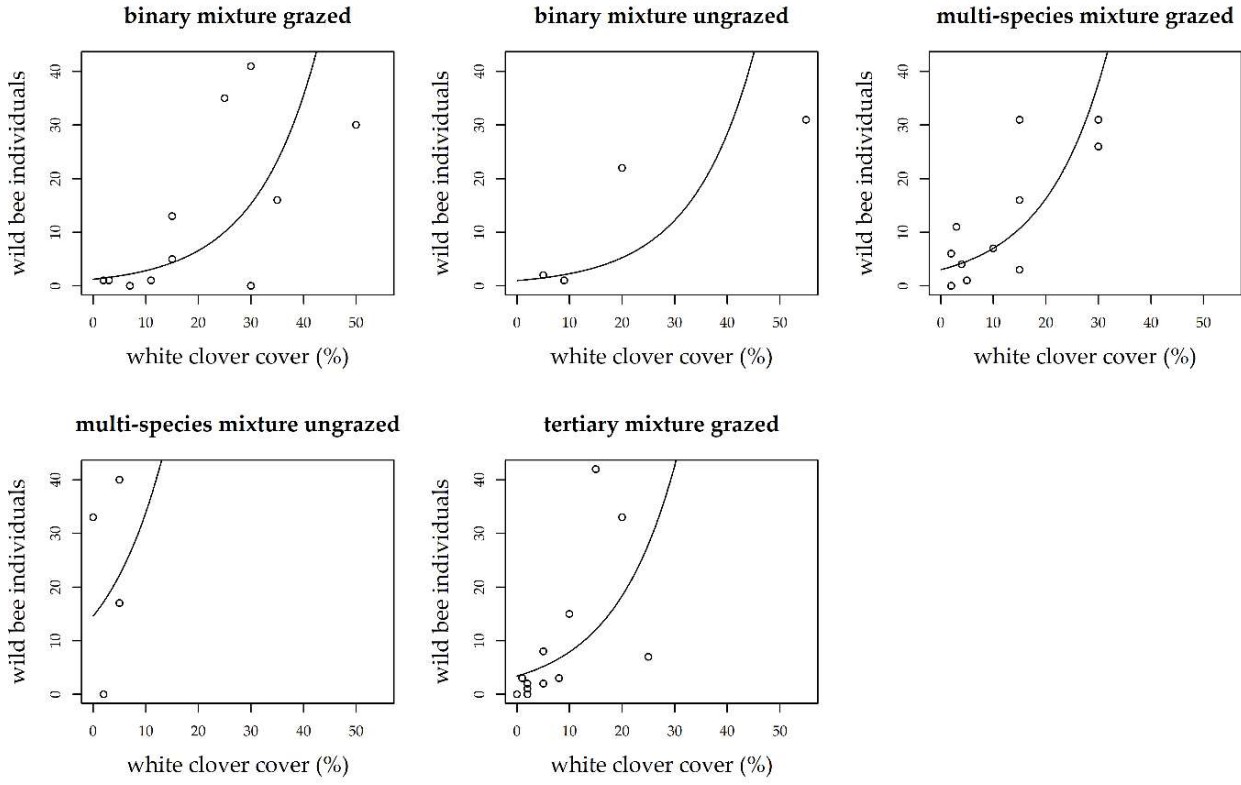

**Figure 1.** Effect of white clover flower cover on wild bee abundance on the mixtures and managements.

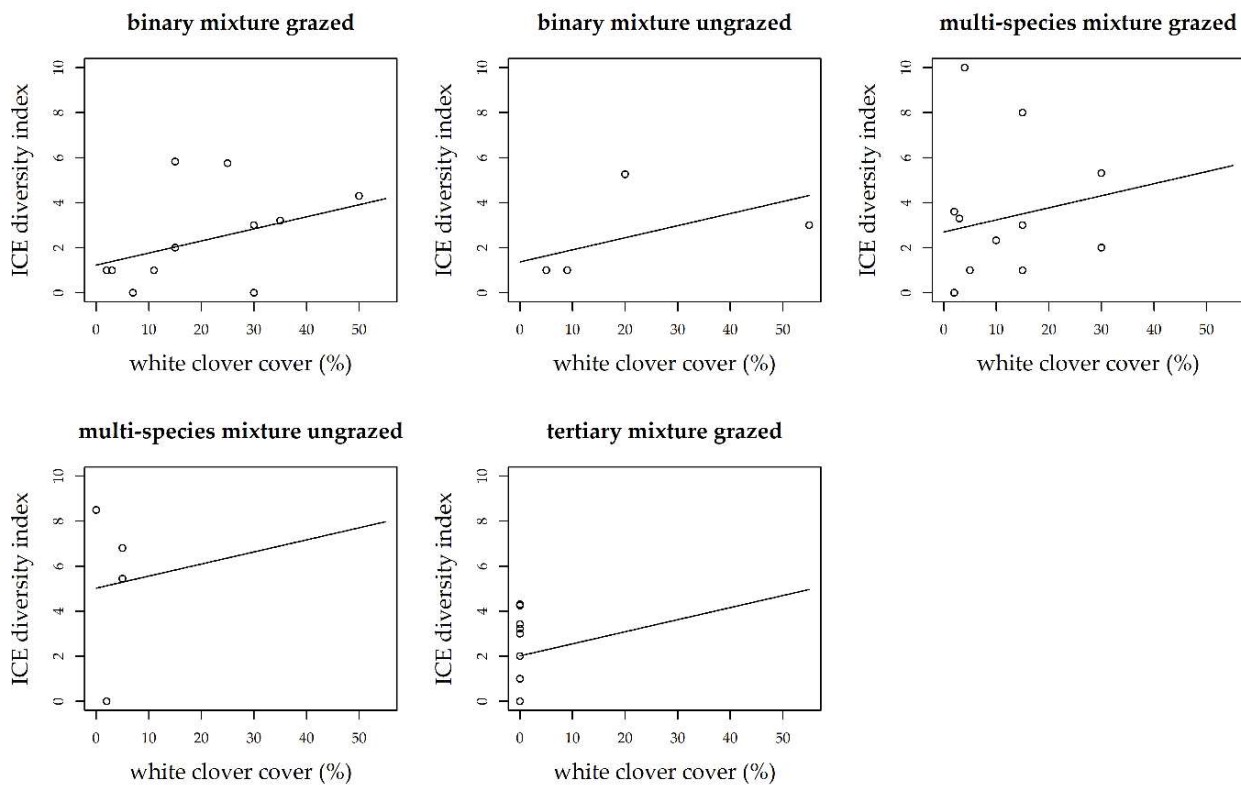

**Figure 2.** Effect of white clover flower cover on ICE diversity index of wild bees on the different mixtures and managements.

### 3.2. Management

In the ungrazed strips of the binary mixture, the total flower cover was 23.5%, white clover was the dominant flowering species with 22.2% flower cover, and 1.3% flower cover of red clover was recorded, which was not a part of the species mixture but migrated from adjacent pastures. The ungrazed strips of the multispecies mixture were dominated by red clover, with a flower cover of 22.8%, followed by birdsfoot trefoil, covering 8%. White clover established a flower cover of 3% and chicory covered 1.3% (total flower cover 35.1%).

The interaction of mixture and management had a significant effect on wild bee abundance ($p < 0.05$). The post hoc analysis revealed that ungrazed strips of the multispecies mixture attracted significantly more wild bee individuals than the ungrazed strips of the binary mixture ($p = 0.008$, Figure 3), and revealed a trend of more wild bees on ungrazed strips of the multispecies mixture when compared with the grazed pasture of the multi-species mixture ($p = 0.053$, Figure 3). On grazed multispecies pastures, we found 11.25 wild bees on average, while we found 22.5 wild bees on the ungrazed multispecies strips. For the binary mixture, we detected no differences in wild bee abundance between the grazed pasture with the ungrazed strips ($p = 0.9713$). The grazing regime did not have an effect on ICE species diversity of wild bees (Table A4). We found 19.6% long-tongued bumblebees on the ungrazed strips of the binary mixture and 22.2% long-tongued bumblebee species on the ungrazed strips of the multispecies mixture.

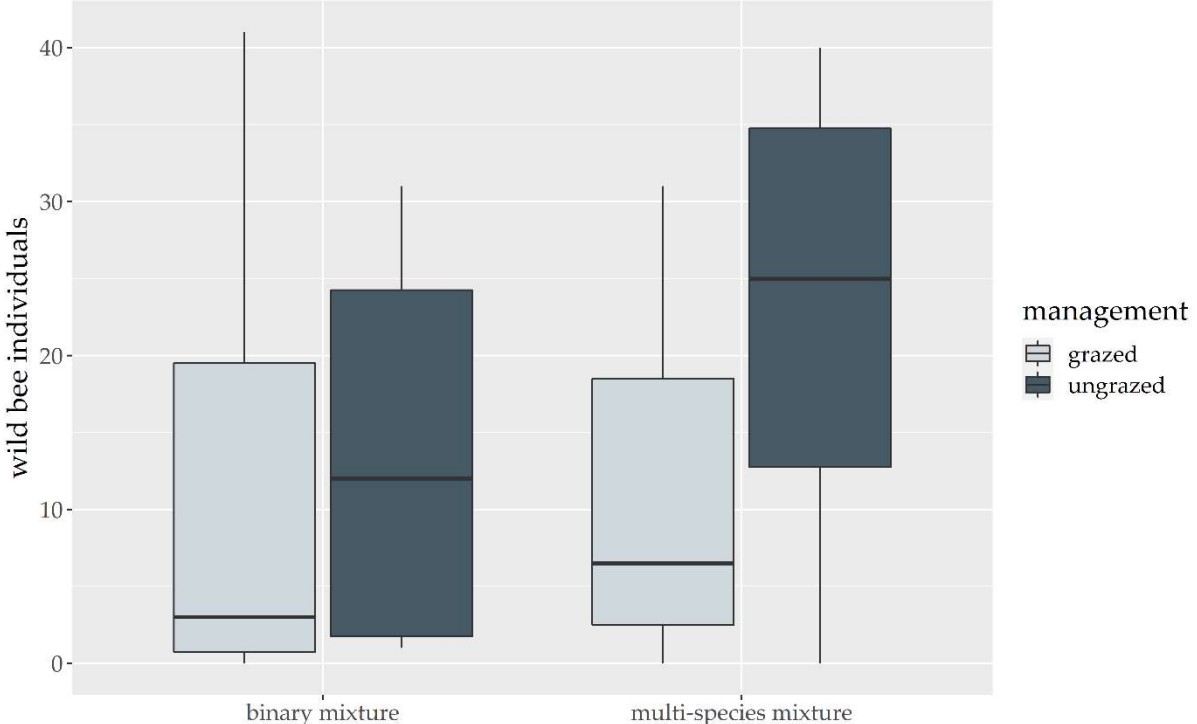

**Figure 3.** Numbers of wild bee individuals on grazed pastures and ungrazed strips (management) of the binary mixture and multispecies mixture.

### 3.3. Yield

The yield was 11.2 tons per hectare and year on the grass-clover pastures of the binary mixture, 13.1 tons on the pastures of the tertiary mixture, and on the multispecies mixture pastures, we measured 12.1 tons. On the ungrazed strips of the binary mixture, we found an average yield of 7.3 tons per hectare, which is approximately 35% less compared with the grazed pastures. On the ungrazed strips of the multispecies mixture, we estimated yields of 10.7 tons per hectare, which is approximately 12% less compared with the grazed pastures. On the intensively managed conventional pastures, we measured yields of 12.5 tons per hectare (Table A5).

### 4. Discussion

The research presented here is part of the superior topic "Eco-efficiency of pasture-based milk production" at Lindhof experimental farm of Kiel University, analysing a wide range of ecosystem services and ecological footprints of low-input dairy farming as a complementary approach to typical highly intensive confinement dairy systems [18,55,56] in Northwest Europe. Compared with high-input confinement milk production systems, based on grass and maize silage, pasturing grass-clover leys results in comparably high milk yield performance per hectare [54]; however, there are very low methane emissions [55], $N_2O$ emissions [68] and nutrient losses via leaching [56], and thus very low carbon and nitrogen footprints per unit of milk produced [18], while enhancing yield stability [69]. Furthermore, combining *L. perenne* and *P. lanceolata*, as in the multispecies mixture in this experiment, enhances soil microbial activity [70]. We started the research presented here with the overall hypothesis that more diversity in botanical composition of ley seed mixtures might also enhance aboveground biodiversity by representing attractive habitats for insects. We can show here that, compared with conventional permanent grassland used for silage production, moderate species enrichment in grass-clover pastures may also benefit bumblebee abundance and multifunctionality in agroecosystems. Our study

also supports previous studies showing that high yields can be achieved with grass-clover mixtures [69].

On conventionally managed grasslands, no flowering resources and no wild bees were recorded; in contrast, 541 wild bees (bumblebees and 1 solitary bee) were recorded in plant-species-enriched grass-clover swards. Wild bee abundance and species richness did not differ among the three grass-clover mixtures, but both increased with the flower cover of white clover. We could also show that the potential of the species-enriched mixtures to promote wild bees is reduced by the intense grazing and that more long-tongued bumblebees were found on strips of the binary mixture and multispecies mixture when grazing was excluded.

Legume mixtures incorporating *Trifolium spp.* can enhance bumblebee abundance and species richness compared with conventionally managed grasslands [52,53]. As wild bee pollination [21,22], especially pollination of bumblebees [52], increases the yield of many crops, bumblebees are desirable in agricultural landscapes. Populations, however, are in decline [44], and grass-clover mixtures can be a tool to ensure the pollination of crops. Yet, such measures should not only focus on closed flowers [44] but also open [24] flowers; therefore, grass-clover mixtures should target more specialized species, and solitary wild bees in general, that are of conservation concern. AESs that target the intrinsic value of biodiversity should be distinguished from AESs that preserve ecosystem service provisioning [71]; grass-clover mixtures should be categorized to the AESs that focus on ecosystem service provisioning.

While intensively managed grasslands with high fertilization and defoliation through grazing or cutting are poor in plant species richness [49,50,72], therefore offering no forage resources for wild bees, the low-input grass-clover pastures in this experiment offered mainly white clover as a foraging resource for bumblebees. Even such modest enhancements of plant diversity, however, may be expected to benefit flower visitors such as bumblebees because intensively used grasslands account for 17.4% of land cover in the EU [73]. Moreover, in rotational stocking systems, flowering resources can be present on a farm-level from May to September. We could also show that the yields of grass-clover mixtures are similar to intensively managed conventional pastures [74], the forage quality is high in both the binary mixture and the multispecies mixture [55], and grazing of the grass-clover mixtures results in higher yields compared with mechanical harvesting [75]. These all imply that the organic ley system with a grass-clover mixture has the potential to create beneficial effects for the environment while maintaining economic benefits.

Here, we showed that bumblebee abundance increased with the flower cover of white clover, independent of the type of grassland mixture. It was also recently found that bumblebee abundance responded to floral density rather than species richness of Fabaceae mixtures in a plot experiment [76]. White clover was important for short- and medium-tongued bees, whereas red clover attracted long-tongued bumblebees [76], a pattern congruent with findings for bumblebees in a landscape-scale study [77]. Bees tend to visit rewarding plant species repeatedly [78], which could likely be the case in this study, since Fabaceae, like white clover, are a high-quality forage resource for bumblebees [79] due to the high amino acid and protein content of the pollen [78]. In accordance with the attractiveness of Fabaceae to flower visiting insects, as shown in this study, the loss of more abundant plant species such as white clover and pollinators such as bumblebees can cause network collapse [80]. In turn, this implies that grass-clover pastures can be an important tool to maintain plant-pollinator networks in agroecosystems, even though white clover and bumblebees are both common species.

While organic farming was shown to support higher bee diversity than conventional farming [81–83], depending on the intensity of management, plant-pollinator networks in organic farming may still remain small, with only minor improvements compared with conventional farming systems [49,84]. Thus, to facilitate more diverse plant-pollinator networks in organic farming systems, a multispecies mixture could be implemented.

Even though the multispecies mixture in this experiment potentially has a higher resource diversity than the binary mixture and tertiary mixture, no difference in wild bee abundance and species richness was observed. This absence was very likely related to the intensive grazing regime which mainly allowed white clover to flower. While white clover did promote bumblebee abundance, species richness of wild bees was not facilitated as no solitary wild bees were caught. Other studies investigating florally enhanced grasslands have shown an increase in species richness in wild bees when the management was less intense [52,53]. Plant species that were part of the multispecies mixture, such as red clover and birdsfoot trefoil, can support rare species, e.g., *Bombus hortorum* [79,85] and *B. muscorum* [86], and in general rare long-tongued bumblebee species [79,87], but also solitary wild bees, such as Osmia, Lasioglossum and Andrena [88].

Within the multispecies mixture, all flowering plant species (white clover, red clover, birdsfoot trefoil) are Fabaceae, despite chicory (Asteraceae), which was highlighted as an important species having both high pollinator visitation and desirable agronomic properties [52] that can potentially promote many Lasioglossum species, e.g., *L. albipes*, *L. calceatum* and *L. morio* and endangered species, e.g., *L. fulvicorne*, *L. nitidulum* and *L. zonulum*, none of which forage on white clover, and can potentially be found in agricultural landscapes of Schleswig-Holstein [88]. The attractiveness of chicory to wild bees could not be confirmed in this experiment. The grazing regime on the pastures was too intense to allow chicory to develop a high number of flower buds and, as a consequence, no wild bees could find foraging resources with open flower buds. On the ungrazed strips of the multispecies mixture, we found chicory, but its flowers are only open from 6 a.m. to 11 a.m. and transect surveys in this study only covered parts of this period. In order to increase the proportion of plants with open flowers to attract wild bee species with shorter tongues than bumblebees, the grass-clover mixture could be extended with, e.g., catsear (*Hypochaeris radicata* L.), common yarrow (*Achillea millefolium* L.) and brown knapweed (*Centaurea jacea* L.). To achieve the full potential of species-enriched grass-clover mixtures in supporting flower-visiting insects, the grazing regime could be adjusted to flowering times of the plants in the grass-clover mixtures, so that, at least on some pastures, plants other than white clover can produce flowers. Furthermore, grass-clover pastures could be lax grazed by young livestock and calves, to decrease grazing pressure on some of the pastures and thus enhance spatial heterogeneity of the sward structure, resulting in a higher share of flowering species that have been rejected by grazing cattle due to ageing and stemmy material, as previously shown on grasslands in Mongolia with lax sheep grazing [89]. The yield measurements, however, show a decrease in yield on the ungrazed strips for both the binary mixture and the multispecies mixture compared with the grazed pastures. In relative terms, this decrease was much smaller for the multispecies mixture than the binary mixture.

On the ungrazed strips of the multispecies mixture, there was a flower cover of 22.8% of red clover and 8% of birdsfoot trefoil, which was substantially higher compared with the grazed pasture, with a flower cover of red clover of 4.3% and a flower cover of birdsfoot trefoil of 0.4%. Consequently, wild bee abundance was twice as high on the ungrazed strips compared with the grazed pastures of the multispecies mixture. Of all bumblebees visiting white clover on ungrazed strips, 1.3% were long-tongued, while for red clover, it was 36.5%, indicating that red clover was an attractor for rare long-tongued bumblebees. Studying a similar species mixture with white clover, red clover and birdsfoot trefoil that was not agriculturally used, a study found congruent results, showing that, of all the bumblebees visiting, 33% were long-tongued [53]. Even though long-tongued bumblebees prefer plant species with longer corolla tubes, such as red clover, they can use white clover as a foraging resource despite the short corolla tube [77]. In this study, we found approximately 20% long-tongued species on ungrazed strips of both the binary mixture and the multispecies mixture, indicating that the exclusion of parts of the pasture from grazing was beneficial for long-tongued species regardless of the mixture. Considering the higher yield level under grazing and a relative decrease in yield of about a third of that in the binary mixture, the multispecies mixture allows the provisioning of both biomass production for high

quality herbages and wild bee conservation without compromising economic benefits for the farmer.

## 5. Conclusions

To counteract the loss of biodiversity and ecosystem functioning related to agricultural land use, finding solutions for ecological intensification in agriculture is crucial. Increasing floral diversity on pastures has potential, because intensively managed grasslands are widespread but lack flower resources. This study indicates that species-enriched grass-clover swards can promote generalist bumblebee abundance; therefore, we concluded that the grass-clover pastures studied here are AESs that promote ecosystem service provisioning. We found that organic grass-clover pastures increase the abundance of generalist bumblebee species in comparison with conventional pastures. We found no effect of increasing floral diversity on the abundance or richness of solitary wild bees, most likely because the dominant flowering plant was white clover, which is known to mainly attract bumblebees. The species richness of wild bees was generally not affected; although, we noted that more rare long-tongued bumblebees were present on the grass-clover mixture if grazing was excluded. In order to fully express their potential to promote rare long-tongued bumblebee species and solitary wild bees, the tested multispecies mixture should be augmented with more species with open flowers and successive phenologies, grazing management should be adjusted to these phenologies, and strips of the pastures should be excluded from grazing for a more continuous provisioning of diverse floral resources.

**Author Contributions:** Conceptualization, H.B., F.T., R.L. and T.D.; methodology, H.B. and T.D.; field work, H.B. and K.L.; insect determination, H.B. and K.L.; data analysis, M.H., H.B. and T.D.; writing—original draft preparation, H.B.; writing—review and editing, T.D., C.K. and F.T.; supervision, T.D. and F.T.; funding acquisition, F.T. All authors have read and agreed to the published version of the manuscript.

**Funding:** H.B. is supported by the Evangelisches Studienwerk Villigst foundation, under the research program: "Third Ways of Feeding the World", and who provided funding in the form of a doctoral scholarship. We acknowledge financial support by Land Schleswig-Holstein within the funding programme Open Access Publikationsfonds.

**Institutional Review Board Statement:** We obtained permits to conduct our fieldwork from the Landesamt für Landwirtschaft, Umwelt und ländliche Räume.

**Data Availability Statement:** The data presented in this study are available on request from the corresponding author.

**Acknowledgments:** The authors would like to thank the anonymous reviewers for valuable comments to improve the quality of this article.

**Conflicts of Interest:** The authors declare no conflict of interest.

## Appendix A

**Table A1.** Composition and sowing rate (kg ha$^{-1}$) of the grass-clover mixtures.

| Species | Variety | Binary Mixture | Tertiary Mixture | Multispecies Mixture |
|---|---|---|---|---|
| Perennial ryegrass (*Lolium perenne*) | Discuss | 6 kg | 6 kg | 4 kg |
| | Calvano1 | 6 kg | 6 kg | 4 kg |
| | Astonenergy | 6 kg | 6 kg | 4 kg |
| | Astonhockey | 6 kg | 6 kg | 4 kg |
| Red clover (*Trifolium repens*) | Vysocan | 2 kg | 1 kg | 0.65 kg |
| | Liflex | 2 kg | 1 kg | 0.65 kg |
| White clover (*Trifolium pratense*) | Harmonie | | 3 kg | 1.5 kg |
| | Larus | | 3 kg | 1.5 kg |
| Plantain (*Plantago lanceolata*) | 4n'Herculese | | | 1 kg |
| Chicory (*Cichorium intybus*) | Spadona | | | 2 kg |
| Sheep's burnet (*Sanguisorba minor*) | Burnet | | | 2 kg |
| Caraway (*Carum carvi*) | Volhouden | | | 2 kg |
| Birdsfoot trefoil (*Lotus corniculatus*) | Lotanova | | | 3 kg |

**Table A2.** Results of the Post hoc test analysing the differences of wild bee abundance of the mixtures.

| | Estimate | SE | z-Value | p-Value |
|---|---|---|---|---|
| binary mix.—tertiary mixture | 1.0248 | 0.5895 | 1.738 | 0.191 |
| binary mix.—multispecies mix. | 0.9044 | 0.5662 | 1.597 | 0.247 |
| tertiary mix.—multispecies mix. | −0.1205 | 0.5602 | −0.215 | 0.975 |

**Table A3.** Results of the Post hoc test analysing the differences of the ICE diversity index of the mixtures.

| | Estimate | SE | z-Value | p-Value |
|---|---|---|---|---|
| tertiary mix.—binary mix. | 0.7870 | 1.8747 | 0.420 | 0.907 |
| multispecies mix.—binary mix. | 1.4718 | 1.8068 | 0.815 | 0.694 |
| multispecies mix.—tert. mix. | 0.6848 | 1.8199 | 0.376 | 0.925 |

**Table A4.** Effect of management (grazed, ungrazed) on the ICE diversity index of wild bees.

| | Estimate | SE | z-Value | p-Value |
|---|---|---|---|---|
| binary mix. ungr.—binary mix. grazed | 0.143 | 2.021 | 0.071 | 1.000 |
| multispecies mix ungr.—multispecies mix. grazed | 2.324 | 2.032 | 1.144 | 0.523 |
| multispecies mix. ungr.—binary mix. ungr. | 3.652 | 2.290 | 1.595 | 0.262 |

**Table A5.** Results of the yield measurements, dry matter in tons per year and the percentages of the total dry matter per plant species.

| | Binary Mixture, Grazed | Tertiary Mixture, Grazed | Multispecies Mixture, Grazed | Binary Mixture, Ungrazed | Multispecies Mixture, Ungrazed | Conventional |
|---|---|---|---|---|---|---|
| Perennial ryegrass | 45.85 | 41.85 | 35,05 | 59.7 | 33.3 | 100 |
| White clover | 53.55 | 27.85 | 27.65 | 39.1 | 13.8 | |
| Red clover | | 29.75 | 19.2 | | 37.8 | |
| Birdsfoot trefoil | | | 2.95 | | 11.1 | |
| Other sown species | | | 14,95 | | 2.9 | |
| Unsown species | 0.6 | 0.55 | 0.35 | 1.2 | 1.1 | |
| Total per year | 11.2 | 13.1 | 12.1 | 7.3 | 10.7 | 12.5 |

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
