# Peer review of "Species-Enriched Grass-Clover Mixtures Can Promote Bumblebee Abundance Compared with Intensively Managed Conventional Pastures"

_agronomy, doi:10.3390/agronomy12051080_

Round 1

Reviewer 1 Report

The paper evaluates the effects of three grass-clover mixtures with varying levels of plant species richness and different management regimes on the abundance and richness of wild bee pollinators.

Results come from one short season (from 25 July to 13 September, on two occasions) even if all plant species grown in mixtures are perennials.

The manuscript is well written with a plenty of references. Sometimes, references are even redundant considering the presented data.

However, no agronomic data are presented and information on mixture components in MM section is too scarce. Therefore, this paper approach (based on bumblebee abundance only) seems too narrow for Agronomy journal, considering the subjected areas of Agronomy MDPI journal.

Vice versa, the same would have been for a paper dealing with agronomic information on mixtures only but submitted to an entomology paper without reporting information on bumblebees.

My general suggestion is that authors should:

  • Include additional quanti-qualitative data on the investigated forage mixtures
  • Based on their findings and conclusions on bumblebees, discuss the related agronomic implications and suggest changes/modifications for improving the management of the investigated mixtures.

Other comments

Materials and methods

Spatial and temporal information on mixtures must be improved.

L 115 please, specify which year instead of in two consecutive years also indicating the month of sowing.

L 116 please, specify which year instead of in the third year; sativa instead of satina.

L 118 please, specify in the spring of which year.

L 120 – 124 Please, mention name of commercial varieties and sowing rate (kg/ha) of the investigated mixture species, which are essential information.

L 127 – 128 please, specify which year correspond to first or second year of production.

L 143 plant species richness and flower cover were recorded in 1m2. Please, explain in detail how these measurements were carried out as well as the units adopted.

L 170 While we found an average flower cover of 18.4%.....

Please, how 18.4% is obtained needs to be explained before

Authors stated at L 359: “Moreover, in rotational stocking systems flowering resources can be present on a farm-level from May to September…..”

If the above statement is correct,

considering at L125-126: “The grass-clover pastures were rotationally stocked every three or four weeks for two to four days from April to September with an average stocking rate of 2.0 livestock units per hectare”,   

why

did you sample from 25 July to 13 September?

Reviewer 2 Report

I’ve hardly any corrections or comments to make for this article. In global terms, everything is correct. I’m only going to point out some details that could improve it, not being faults in themselves or a problem for their understanding if they are left as they are now. Rather they are suggestions.

On lines 68-69 one hopes to find the objectives to be achieved after having argued the introduction of the topic. It would be convenient to move this phrase to the beginning of line 101.

On lines 187 to 190, when talking about long-tongued bee species, it’s assumed that they are all bumblebees, except for one Andrena captured in this trial, the rest of the captures were Bombus and therefore all long-tongued species. As it’s written, it is cumbersome and confusing. The same happens on lines 213 and 214.

In general, two ways of referencing in the same text are not usually used, that is: “According to Hall et al. 2005” and “[58]”. It is better to choose one mode and stick to it throughout the text. This occurs on lines 249, 262 and 267.

In the specific case of the “Scheper et al. 2013”, line 249, I can't find the reference in the references section.

On line 290 and 291, there’s a data redundancy and it’s not necessary. Putting only “…Cichorium intybus L. (Asteraceae) was highlighted…” thus making it clear to which family the plant belongs to.

On line 529, change SCHALL to Schall.

On line 549, delete “;” behind Vol. 2, because it gives the feeling that information is missing.

Lastly, I’m not sure if in lines 549, 544 and 534, the years of the references have been left out in bold on purpose or it’s a misprint.

Reviewer 3 Report

The authors propose a manuscript titled “Plant resources utilization among different ethnic groups of Ladakh in Trans Himalayan Region”. The authors consider a very interesting and original topic about the nomadic pastoral indigenous communities of Ladakhi people share roots with Tibetan culture in terms of food, clothing, religion, festivals, and habits, and rely widely on plant resources for survival and livelihood. The survey was conducted don plant resources of the Balti, Beda and Brokpa communities of the Ladakh region, Trans Himalayas by open and close-ended semi-structured interviews and group discussions for a total of 105 plant species belonging to 82 genera and 39 families used for services such as medicine, fuel wood, fragrance, oil, food, flavor, fodder, decoration, and dye and the medicinal use was most prevalent followed by fodder and fuel wood, also the leaves were the most preferred plant part used, followed by roots and flowers. Also the authors give new data for some species that played a significant role in the cultural and religious ritual aspects, whereas other species were commonly used as a livelihood source among Ladakhi communities alleviation. The manuscript is original in the data compared to other similar articles and is able to be published on international audience. However I believe it is necessary to implement the manuscript with few formal concepts that the authors will have no problem to accepting as they are designed to improve the work.

Title

The title of the manuscript and the content of the abstract do not coincide. Please adapt the title to the abstract

Abstract

Please summarize without dividing into 4 sections

Introduction

Please correct in the suggested way

  • Lines 60-61. Please add reference for this statement: “The growing demand for food worldwide has fundamentally changed agricultural systems since the 1950s [Add reference]”
  • Lines 63-66. This intensification is driving spatial [8–10] and temporal [11] homogenization of agricultural landscapes which imposes a serious threat to biodiversity and ecosystem functioning [12]. Thus, reintroducing and conservation of plant biodiversity is considered to be key in promoting agro-biodiversity [13,14], especially for Crop Wild Relatives (CWRs) that can crossing with the cultivated relatives [Perrino and Wagensommer 2022], favoring many natural processes useful for maintaining a high biodiversity. . In recent years, measures towards ecological intensification in agriculture have been discussed and promoted [15–17. add Perrino et al. 2014].

References to be added:

  • Perrino, E.V.; Wagensommer, R.P. Crop Wild Relatives (CWRs) Threatened and Endemic to Italy: Urgent Actions for Protection and Use. Biology 2022, 11, 193. https://doi.org/10.3390/biology11020193
  • Perrino, E.V.; Ladisa, G.; Calabrese, G. Flora and plant genetic resources of ancient olive groves of Apulia (southern Italy). Genetic Resource and Crop Evolution 2014, 61, 23-53. Doi: 10.1007/s10722-013-0013-1

Materials and methods

Some observations

  • Lines 111-112. Please specify the geographic system used (WGS84?) for this statement: The study took place in 2019 at the Lindhof experimental farm of Kiel University, in Schleswig-Holstein, Germany (54° 28’N; 9° 58’E);
  • Lines 113-114. Please specify in which period range the authors want to refer on temperature and precipitation. “The mean annual temperature of the study area is 8.7 °C with a mean annual precipitation of 785 mm”;
  • Line 116. Avena satina or Avena sativa?. Please check;
  • Line 123. Plantago lanceolata instead Plantago lanceolate

Results and discussion

Few observations. The figures and tables are clears.

  • Table 1. Pay attention the name of the author of the scientific name always not in italic. E.g. Arnebia guttata Check whole table.
  • Figure 3a seems distorted
  • Lines 222-223. Add a reference for the following statemnet “Furthermore, according to various researchers [49,50,51, Valerio et al. 2021], the members of these families have a high content of useful bioactive compounds”;
  • Line 145. Please specify better the method used including which software
  • Line 242. Trifolium pl. instead Trifolium spp.
  • Lines 290-291. Chicorium intybus (Asteraceae) instead Cichorium intybus (L.; Asteraceae)

Conclusion

Some more information on prospective studies

Reference

Please see the guidelines of the jou

Round 2

Reviewer 1 Report

Dear Authors,

thanks for the given explanations.

Revised version of paper has been improved by inserting specific details, which were missing mainly in MM section, as well as basic productive data on the investigated mixtures.

The manuscript also supplies a plenty of useful information.

Author Response

Dear Reviewer,

Thank you again for your second review report. We are glad you found our changes helpful. As you did not suggest any further changes, we did not do any more alterations.

Reviewer 3 Report

Dear authors, I appreciate the work done. I don’t know why I have received a old version of your manuscript with different title. 
In anyway, this last version is able to be published.

congratulation,

reviewer

Author Response

Dear Reviewer,

Thank you again for helping with the submission of our manuscript. We are glad you found our changes helpful. We did not make further changes, as you did not suggest any improvements.